# AlphaIntegrator: Transformer Action Search for Symbolic Integration Proofs

## Abstract

We present the first correct-by-construction learning-based system for step-by-step mathematical integration. The key idea is to learn a policy, represented by a GPT transformer model, which guides the search for the right mathematical integration rule, to be carried out by a symbolic solver. Concretely, we introduce a symbolic engine with axiomatically correct actions on mathematical expressions, as well as the first dataset for step-by-step integration. Our GPT-style transformer model, trained on this synthetic data, demonstrates strong generalization by surpassing its own data generator in accuracy and efficiency, using 50% fewer search steps. Our experimental results with SoTA LLMs also demonstrate that the standard approach of fine-tuning LLMs on a set of question-answer pairs is insufficient for solving this mathematical task. This motivates the importance of discovering creative methods for combining LLMs with symbolic reasoning engines, of which our work is an instance.

## 1 Introduction

Large language models (LLMs) based on the transformer architecture (Vaswani et al., 2023) have demonstrated remarkable abilities across diverse tasks, such as language translation, code generation, and engaging human-like conversations (OpenAI, 2024). However, applying these models to mathematics presents significant challenges. Their autoregressive nature makes them prone to hallucinations and errors during inference. Advancements such as Chain-of-Thought (CoT), self-consistency, and process supervision help generate more accurate multi-step reasoning (Wei et al., 2023), (Wang et al., 2023), (Lightman et al., 2023). However, unlike general language tasks, mathematics demands absolute rigor and precision, where even minor errors are unacceptable. Mathematical correctness relies on faultless execution of logical steps and computations. LLMs often fail to achieve this consistently and there is no provable method which ensures the correctness of their mathematical reasoning.

**Mathematical Integration**   A fundamental mathematical task is one of indefinite integration of mathematical expressions, a problem with no straightforward algorithmic solution. Existing methods for solving this task fall into two categories: those that directly output the antiderivative, and those that provide step-by-step proofs.

Lample and Charton (2019) proposed a learning-based approach, training a seq2seq model to generate the antiderivative directly, without steps and with correctness verification left as a separate problem. Welleck et al. (2021) demonstrated that such models do not generalize well, even though they might have high test accuracy, as the neural network needs to mechanically learn how to carry out complex operations like applying the partial fractions algorithm or dividing two large numbers. The algorithm proposed by Risch (1969) reduces the integration problem into finding poles of certain algebraic functions. Risch's method is pseudo-complete for indefinite integration, but it only applies to the restricted setting of functions with elementary antiderivatives and similarly does not produce intermediate steps. The full description of the method is longer than 100 pages and has never been fully implemented. Current symbolic solvers often include a simplified, heuristic version. However, the resulting answers, while always correct, are not very illuminating. Further, in contrast to learning-based approaches, this method does not directly generalize to similar tasks, such as non-elementary or multidimensional integration, or general theorem proving.

To generate step-by-step proofs, SymPy's *manualintegrate* module can be used, which applies various heuristic techniques recursively to construct a solution. However, this method is slow and prone to failure on simple integrals due to its reliance on manual pattern matching. Another option is leveraging state-of-the-art language models like GPT-4. However, these models lack guarantees of correctness. Our experiments further show that, despite their vast training data and billions of parameters, such models often perform poorly on complex integrals.

**Our Work: correct-by-construction learning-based integration**   In this work we introduce the first open system which combines the strengths of both symbolic engines and GPT transformer models in order to learn to integrate mathematical expressions in a step-by-step, provable manner. Our approach is inspired by the groundbreaking advancements of AlphaProof and AlphaGeometry (Trinh et al., 2024), where language models interact with a symbolic engine to generate a solution that is guaranteed to be correct. Concretely, we designed a novel symbolic engine and generated synthetic data used to train a GPT transformer language model capable of sophisticated interaction with this engine.

**Main contributions**   Our key contributions are:

- The first dataset for rigorous step-by-step derivation proofs for indefinite integration.
- A versatile open-source symbolic engine to interact with mathematical expressions through axiomatically correct actions with a novel encoding.
- A (very small) transformer model which surpasses in performance the leading open-source step-by-step integration baseline. Our evaluation also demonstrates that our tool can effectively guide search in a complicated action space and thus surpass its own dataset generator in both completeness and efficiency, through strong generalization.

The rest of the paper is organized as follows. In Sections 2 and 3, we introduce the symbolic engine and our representation of mathematical expressions. In Sections 4 and 5, we explain how to generate synthetic data for integration and how we train the model. Finally, we explain how we run and evaluate the model in Sections 6 and 7.

## 2 SYMBOLIC ENGINE WITH PARAMETRIC ACTION SPACE

We depart from the typical approach of solving mathematics problems with LLMs done by fine-tuning on question-answer pairs (Shao et al., 2024; Yang et al., 2024). Instead, our language model interacts with a symbolic engine exposing a parametric action space. This guarantees that every step taken by the model is correct (or null) since rewrites of mathematical expressions are permitted only through the symbolic engine.

In each step, the symbolic engine takes in a mathematical expression $f$, a subexpression $g$, an action $a$, and action parameters $p_1, \ldots, p_n$, if applicable. It returns an expression that results from applying action $a(p_1, \ldots, p_n)$ to subexpression $g$, along with a boolean that specifies whether the expression was modified or not. The expression is not modified if $g$ is not a valid subexpression of $f$, or if the action is not valid on this subexpression. Figure 1 shows an example of a successfully executed action. A full list of actions can be found in Table 4 in Appendix A.1.

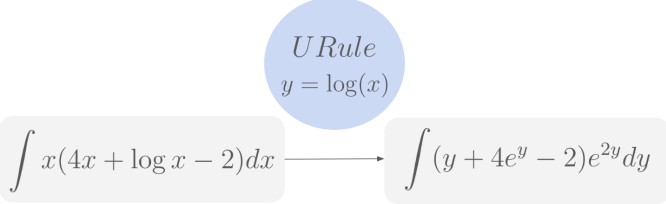

Figure 1: The symbolic engine takes in the subexpression $\int x(4x + \log x - 2)dx$ and applies the substitution rule with $y = \log x$. The symbolic engine will realize the rule by first differentiating $y(x)$ and dividing the integrand by $y'$. Then, it will substitute by $y$ wherever it observes $y(x)$ and then solve for $x = g(y)$ to substitute the remaining terms.

The symbolic engine holds in its state a dictionary of pairs of expressions encountered and the changes of variables that are active in the respective expression. The dictionary is used to backtrack whenever we reach again a state that has already been explored. We undo via backsubstitution any changes of variables for which there are no integrals remaining with a substituted variable.

Note that this is parallel to theorem proving by interacting with a formal language (Xin et al., 2024; Polu and Sutskever, 2020; Lample et al., 2022), which makes our synthetic-data-based approach applicable to a variety of tasks.

# 3 REPRESENTATION OF MATHEMATICAL EXPRESSIONS AND THEOREMS

We now discuss how we represent mathematical expressions and theorems in our symbolic engine, and how we encode these as sequences for interaction with a sequence model.

We represent mathematical expressions as trees. Leaf nodes are number constants or variables, such as $2$, $\pi$, or $x$. Internal nodes are operators and functions, such as $+$ or $\cosh$. We show representations of the expressions $\frac{1}{x+3} + 2\cosh^2(x)$ and $\int x^2 e^x dx$ in Figure 2.

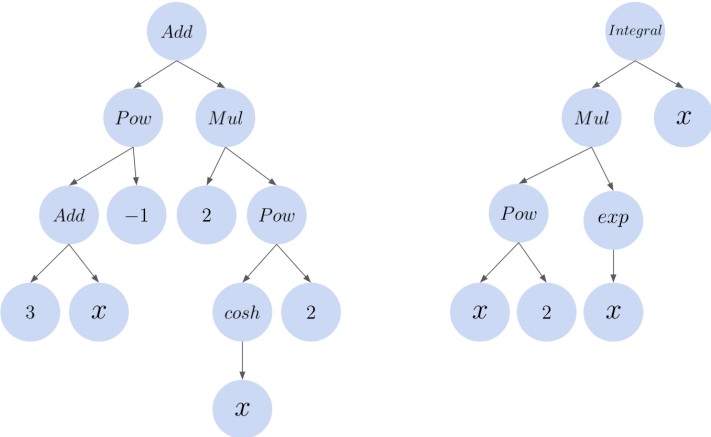

Figure 2: Tree representations of the expressions $\frac{1}{x+3} + 2\cosh^2(x)$ and $\int x^2 e^x dx$

We assume that each operand/function in the tree has a fixed arity. For example, addition and multiplication have two children, while functions like $\sin$ have one. Note that there is a strict ordering of the children. For example, for the *Pow* node, the first child is the base and the second child is the exponent. Some of the operators are symmetric: for example $a + b = b + a$. In these cases, we resort to an arbitrary canonical ordering of the children.

## 3.1 TREE TO SEQUENCE EQUIVALENCE AND PARSING

We turn mathematical expression trees into sequences of tokens in order to process them using transformers. In the following, we show algorithms to construct a one-to-one correspondence between expression trees and sequences under the assumptions above.

**Tree to Sequence**  In order to turn a tree into a sequence, we define a recursive function:

$$treetoseq(v) = \begin{cases} [v] & v \text{ is a leaf node} \\ [v] + treetoseq(c_1) + \ldots + treetoseq(c_{n_{child}}) & \text{otherwise} \end{cases}$$

where $n_{child}$ denotes the number of children of $v$, $c_i$ denotes the $i$-th child, and $+$ denotes concatenation of sequences. This algorithm corresponds to doing depth-first traversal by picking the first child and writing down all the observed values in order. Running *treetoseq(r)* on the root node $r$ of an expression $f$ produces a sequence suitable for transformer tokenization.

**Sequence to Tree**  To parse outputs of the model, we need a function that can unambiguously map from sequences to trees. We achieve this using Algorithm 1, discussed in Appendix B.1. Note that this function returns a tree as well as the remaining part of the sequence. We assert that $remaining$ has to be an emtpy list for the sequential representation to be valid.

## 3.2  TOKENIZATION

We tokenize with unique tokens typical operations such as addition, power, multiplication, as well as all trigonometric, hyperbolic, and special functions (e.g. $erf$). We create 7 symbols (e.g. x,y, etc.) that can be used as variables of integration and change of variables, and we tokenize special constants such as $e, \pi$, and $i$ with their unique tokens. We represent integrals with a special token `INTEGRAL` and rational numbers with a token `RATIONAL` followed by two integers. We tokenize integers using their base-ten representation preceded by a token `INT+` or `INT-`, indicating whether the integer is positive or negative. For simplicity, we do not have a dedicated representation of decimal numbers. For example, the expression $\int(\frac{1}{x+3} + \cosh^2(x))dx$ would be tokenized as follows:

```
INTEGRAL + POW + INT+ 3 x INT- 1 * INT+ 2 POW cosh x INT+ 2 x
```

Finally, we designate a token to each theorem in the symbolic engine. This allows us to distill mathematical expressions and theorems into an exceptionally compact formal language, achieving a minimalist yet expressive vocabulary of just $d_{vocab} = 128$ tokens.

## 4  GENERATING A SYNTHETIC MATHEMATICAL INTEGRATION DATASET

We will train a model to derive step-by-step integrals by predicting single-step rule applications. To this end, we generate fully synthetic step-by-step integration data, described in this section. Note that such a rigorous step-by-step integration dataset, based on a well-defined space of possible actions and on such a broad variety of mathematical expression data, was lacking prior to this work.

### 4.1  RANDOM MATHEMATICAL EXPRESSIONS

To create a large-scale dataset of mathematical expressions, we adopt an algorithm from Lample and Charton (2019), which samples random unary-binary trees and fills the nodes with operators and operands. We generate ∼5M unique expressions with this algorithm described in Appendix C.1.

### 4.2  STEPS OF INTEGRATION

Once we have generated a dataset of random mathematical expressions, we pass them through the *manualintegrate* module of SymPy (Meurer et al., 2017), in order to get a step-by-step solution. Then, we map the solution into a sequence of actions and parameters in our symbolic engine. This results in a sequence of tuples of expression, subexpression, action, and action parameter for each expression. An example of a full solution in this format, generated by our model, is given in Figure 4. There are, of course, many expressions SymPy cannot integrate as it enumeratively tries heuristic methods. Our hypothesis is that transformers are able to generalize to cases not covered by SymPy.

#### 4.2.1  DATA AUGMENTATION WITH INTEGRATION BY PARTS

Let $\Phi, \Psi$ be two random functions generated by the method above, with derivatives $\phi, \psi$. By the rule of integration by parts, we have

$$\int \Phi(x)\psi(x)dx = \Phi(x)\Psi(x) - \int \phi(x)\Psi(x)dx$$

Then, if we know a step-by-step integration of $\phi(x)\Psi(x)$, we can find the steps for $\Phi(x)\psi(x)$ by applying integration by parts with the right parameters and applying the steps of $\phi(x)\Psi(x)$ to the relevant subexpression of the resulting expression. We augment our dataset with this technique by searching for such instances in the previous dataset. Our final dataset consists of 42.5M integration steps and we report further statistics in Table 5.

## 5 MODEL ARCHITECTURE AND TRAINING OBJECTIVE

In this section, we describe our transformer model architecture and the objective we used for training.

### 5.1 ARCHITECTURE AND HYPERPARAMETERS

We use a decoder-only transformer architecture with 6 layers of multi-head attention with 6 heads and a hidden dimension of 384 (Radford et al., 2019). This results in a tiny model with only 10M parameters. We use the AdamW optimizer with $\beta_1 = 0.9$, $\beta_2 = 0.99$, dropout of $\beta = 0.2$, and weight decay of $\lambda = 0.1$ (Loshchilov and Hutter, 2017). We decay the learning rate linearly from $10^{-3}$ to $10^{-4}$ throughout training, with batch size 256. We use a single A100 GPU for training. We choose this simple setting as we observed no performance improvements with larger architectures.

### 5.2 TRAINING OBJECTIVE

We would like our model to propose a subexpression, action, and parameters of the action given a mathematical expression to integrate. Then, the model will be repeatedly fed back with the result obtained through the symbolic engine to find solutions for new expressions. To train our model for this, we shuffle all integration steps into lines structured as follows:

```
START [EXPR] SUBEXPR [SUBEXPR] RULE [RULE] PARAM [PARAM] END
```

Here, terms in parentheses are tokenized mathematical expressions (e.g. $x^2 + \sinh(x)$) or actions (e.g. *PartsRule*). We use the standard Cross Entropy Loss objective, where the model predicts extensions of `START [EXPR] SUBEXPR`. An example from the training dataset looks as follows:

```
START Integral cos + E + x tan INT+ 2 x SUBEXPR Integral cos + E +
x tan INT+ 2 x RULE URule PARAM1 y PARAM2 + E + x tan INT+ 2 END
```

This step corresponds to transforming the integral $\int \cos(x + \tan(2) + e)dx$ using change of variables $y = x + \tan(2) + e$. Applying this with the symbolic engine would result in the integral $\int \cos(y)dy$ while storing the change of variable $y = x + \tan(2) + e$ in its memory.

## 6 ACTION SEARCH

In this section, we describe how we run our model, interacting with the symbolic engine, to solve an integral.

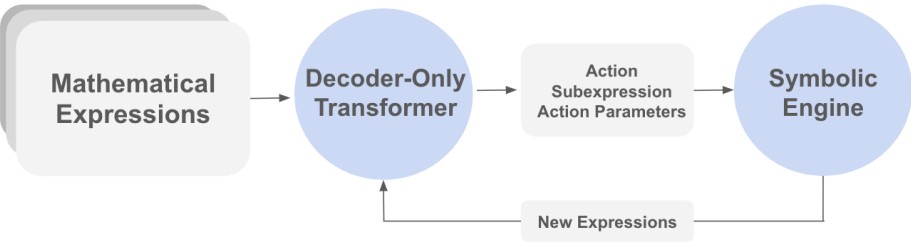

Figure 3: Inference Loop for Integration

Given an expression $f$, we tokenize it and feed it to the transformer. We decode the transformer using beam search with $N$ candidates until we reach the `END` token, i.e., we heuristically search for the sequence with maximum log-probability (Freitag and Al-Onaizan, 2017). Then, we find all valid generated actions (with parameters). We independently run them on the current expression using the symbolic engine. We obtain new expressions, ordered by decreasing log-probability of the proposed action that generated them. We greedily explore the tree resulting from feeding the expressions back into the transformer, until the integral sign disappears. This loop is illustrated in Figure 3.

We tried other decoding techniques, such as top-k and nucleus sampling, but we did not observe any significant improvements in performance (Holtzman et al., 2019). Note that nothing prevents the

model from generating invalid mathematical expressions or invalid actions. However, we observed that the model almost always outputs valid expressions and actions, so we simply discard proposed actions whenever they are invalid.

# 7 EXPERIMENTAL EVALUATION

We explore a number of different directions to evaluate our model. We aim to demonstrate that our method not only significantly outperforms existing step-by-step benchmarks, but is also more efficient and generalizes well, unlike existing learning-based approaches. To illustrate the capability of our model, we present an example solution that it generated, in Figure 4. For this example, SymPy failed to figure out that we can apply the substitution $u = \sin(2x)$. We show further examples in Appendix D.

$$\int \left( 1 + \frac{2\cos(2x)}{\sqrt{\sin^2(2x)+1}} \right) dx$$

AddRule on entire expression

$$\int 1\, dx + \int \frac{2\cos(2x)}{\sqrt{\sin^2(2x)+1}}\, dx$$

Apply ConstantRule on $\int 1\, dx$

$$x + \int \frac{2\cos(2x)}{\sqrt{\sin^2(2x)+1}}\, dx$$

Apply URule $u = \sin(2x)$
on the integral part

$$x + \int \frac{du}{\sqrt{u^2+1}}$$

Apply ArcSinh rule for $\int \frac{du}{\sqrt{u^2+1}}$

$$x + \sinh^{-1} u + C$$

Substitute $u = \sin(2x)$ back

$$x + \sinh^{-1}(\sin(2x)) + C$$

Figure 4: Step-by-step solution generated by AlphaIntegrator.

## 7.1 TEST SET ACCURACY

We hold out a test set of 10k expressions with integration steps, unseen during training. We compare our model, SymPy, and GPT-4o-mini. We run SymPy and our own model with timeouts of 120 and 10 seconds, respectively. We use $N = 5$ for beam search decoding. We observe that this beam search results in correct steps in the proposed actions for $> 99\%$ of the test set, on a step-by-step level. More precisely, this means that if we predict a single step, there is almost always an exact match of the step in the test set within one of the five recommendations resulting from beam search. We prompt GPT-4o-mini with zero-shot CoT and we only check correctness of the result, ignoring any wrong intermediate steps, for a random subset of 1000 expressions. We present accuracy results in Table 1.

Our model significantly outperforms both SymPy and GPT-4o-mini, a state-of-the-art language model. Of particular interest is that our model generalizes beyond its data generator SymPy. We observe that GPT-4o-mini generally fails on long chains of computations, or when expressions are not sufficiently

Table 1: Comparison of model accuracies on the integration task.

| Model | Accuracy | Error Margin |
|---|---|---|
| **AlphaIntegrator** | **87.3%** | $\pm 0.3\%$ |
| SymPy | 83.3% | $\pm 0.4\%$ |
| GPT-4o-mini | 65.5% | $\pm 1.5\%$ |

similar to what is typical. During manual inspection, we observed that the model usually has the right methods in mind, but then cannot execute operations accurately. Typical mistakes include sign errors while doing integration-by-parts or simple errors in arithmetic. For SymPy, it is more often the case that the system is unable to find solutions rather than making mistakes, as some cases are missed by the software. We explore this further in Section 7.4, where we show some of the bugs and limitations we found in SymPy.

## 7.2 EFFICIENT TREE EXPLORATION

To understand how well the transformer model guides us in the tree search, we measure the number of tree nodes explored during integration for both SymPy and our model, on the test set. We find that on average, our model explores $N_t = 12.9$ nodes for each successful integration whereas SymPy explores $N_s = 25.6$. This demonstrates that our model is not only more powerful but also more efficient, as it explores roughly $50\%$ fewer nodes to find solutions. We present the full distribution in Figure 5.

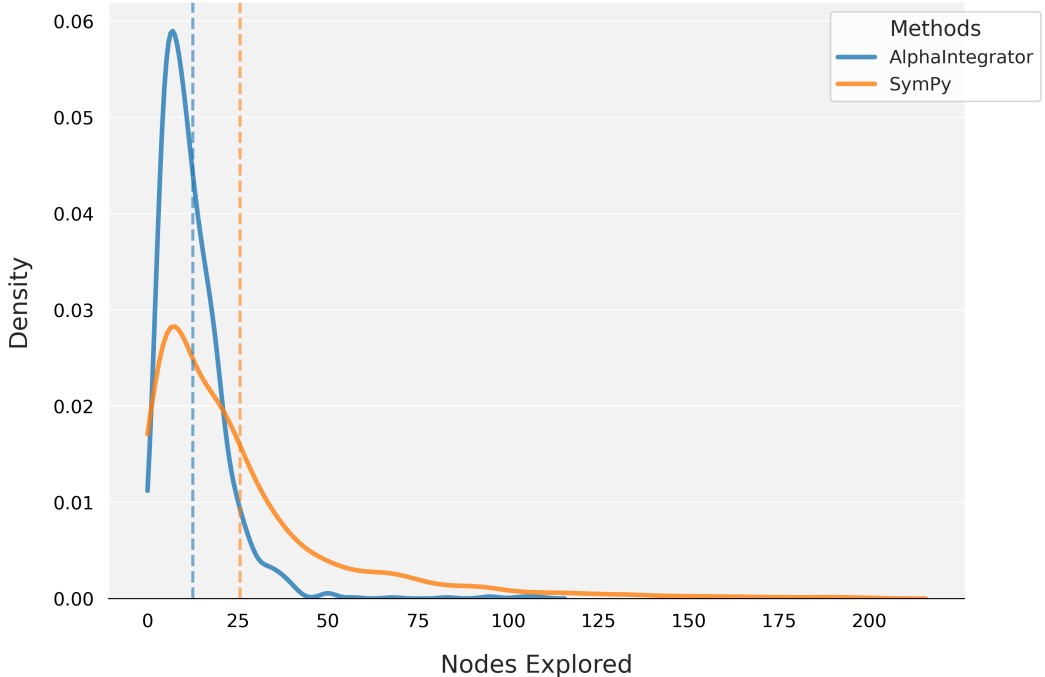

Figure 5: Distribution of number of nodes explored in tree search for AlphaIntegrator vs SymPy.

## 7.3 ROBUSTNESS AGAINST SYMBOLIC BRITTLENESS

Recently, Welleck et al. (2021) introduced the concept of 'symbolic brittleness', where they investigate the seq2seq model by Lample and Charton (2019) that directly predicts the antiderivative of an expression. Their findings showed that while the model performs well on in-distribution problems, it

struggles with robustness to small input variations, fails to generalize compositional patterns, and exhibits poor out-of-distribution performance.

In particular, they manually search for perturbations of expressions that the model can integrate to test its robustness. They introduce the metric *Fail@N*, defined as the percentage of times a model fails to find the correct integral within its top N predictions, as determined by beam search. For instance, Fail@1 represents the failure rate when only the top prediction is considered, while Fail@50 reflects the failure rate when the top 50 predictions are evaluated. For our model, we similarly define Fail@N as the percentage of failures using a beam search of size $N$.

We test both models by perturbing the set of functions that includes sin, cos, exp, tan, by multiplying their arguments and return values by random integers. We report the results in Table 2. The experiments show that AlphaIntegrator is significantly less brittle in almost all cases.

Table 2: Robustness results with simple primitives (top) and validation problems (bottom). Coefficients are sampled from [1, 50].

| Test | Fail@10 Seq2Seq | Fail@5 AlphaIntegrator |
|---|---|---|
| $k_1 \ln(k_2 x)$ | 0.0 | 0.0 |
| $k_1 x$ | 0.0 | 0.0 |
| $k_1 x^{42}$ | 6.1 | 0.4 |
| $k_1 \exp(k_2 x)$ | 20.8 | 6.2 |
| $k_1 \sin(k_2 x)$ | 19.6 | 0.2 |
| $k_1 \cos(k_2 x)$ | 20.7 | 0.0 |
| $k_1 \tan(k_2 x)$ | 17.4 | 14.1 |
| $\frac{1}{k} \cdot f$ | 12.0 | 0.2 |
| $k \cdot f$ | 5.8 | 0.1 |

Welleck et al. (2021) claim that the seq2seq model learns how to copy patterns, however, does not generalize well in primitives that require dividing coefficients (e.g. $\int k_1 \cos(k_2) x \, dx = \frac{k_1}{k_2} \sin(k_2 x)$). Our model does not need to learn how to perform arithmetic operations, but rather has to recognize the pattern, copy a subexpression, and choose what action to apply. We believe that this is the main reason why we manage to obtain a more robust model than direct antiderivative prediction by Lample and Charton (2019). We also note that our method is more general in the sense that it can be applied to any step-by-step computation task with no way of verifying the correctness of the result.

## 7.4 EXPLORING BUGS IN SYMPY

As our method generalizes over SymPy's module, studying examples where SymPy fails and our method succeeds is very useful to find bugs or limitations in the heuristic of the symbolic solver software. Following this method, we found simple bugs in SymPy and reported them as GitHub issues. We show examples of such bugs and failure modes in Table 3.

Table 3: Bugs/Failure Modes in Sympy

| Expression | Description |
|---|---|
| $\int \frac{1}{\cos(x)} dx$ | Not covered by the enumerative search for simple trig. integrals. |
| $\int x \cosh(x) dx$ | Heuristic that does integration by parts fails to try this case. |
| $\int \cos(2x) \tan(x) dx$ | Manages to rewrite $\cos(2x) = \cos^2(x) - \sin^2(x)$ and integrates the second term but fails to rewrite $\tan(x) = \frac{\sin(x)}{\cos(x)}$ to obtain $\int \sin(x) \cos(x) dx$ for simple integration by parts. |

## 8 RELATED WORK

**Deep Learning for Theorem Proving** Deep learning applied to theorem proving has seen significant advancements, particularly in premise selection and proof guidance. Early work such as DeepMath

used CNNs and RNNs to predict the relevance of premises for proving conjectures (Alemi et al., 2016). Later, various papers leveraged decoder-only transformer architectures or pre-trained LLMs to guide proofs through recommending premises and next steps (Polu and Sutskever, 2020) (Yang and Deng, 2019) (Song et al., 2024). These models usually interact with Interactive Theorem Provers (ITPs) such as Lean, Isabelle, or Coq. This usually requires large-scale pre-training data to be successful. On the contrary, we develop a compact language for interaction between the symbolic engine and generative model. This allows us to easily generate synthetic data and circumvent the pre-training required to learn the syntax of the language.

**Tree Structured Neural Networks** A recent body of work has employed neural network structures that have inductive biases for processing tree-structured inputs which is the case when dealing with mathematical expressions or abstract syntax trees. For example, Tai et al. (2015) proposes TreeLSTM, a generalization of LSTMs to tree-structured network topologies. (Huang et al., 2018) uses this architecture to train baseline models for the formalization of the Feit-Thompson Theorem. (Arabshahi et al., 2018) trains the same architecture to model mathematical equations and verify their correctness. In this work, we focus on decoder-only architectures, as they have become more standard, and we aim to build a system that is not only efficient but also easily transferable to other tasks. This allows for greater flexibility and adaptability across various problem domains.

## 9 CONCLUSION AND FUTURE WORK

We introduced a novel approach for step-by-step integration using a transformer model guided by a custom symbolic engine. The policy captured by the transformer model is learned from a synthetically generated dataset of integration rules. A major advantage of our work is that it guarantees the final expression is always sound. This follows from the fact that the policy always applies a correct-by-construction integration rule, realized by the symbolic solver. Our experimental evaluation demonstrates strong generalization, surpassing its data generator in accuracy and efficiency. Interestingly, it also provides insights into potential errors found in modern heuristic solvers. We demonstrated significant improvements compared to direct approaches using LLMs, which typically fine-tune an LLM on a dataset of question-answer pairs. We exhibit better robustness compared to other learning-based approaches and our method naturally generalizes to other settings, where results without intermediate steps may be hard to verify. Limitations include reliance on synthetic data and the scope of integration techniques that we handle. Future work will focus on more general approaches to training such as reinforcement learning, expanding the action space, and extending the model to other mathematical tasks.

## 10 ETHICS STATEMENT

This work focuses on improving symbolic integration using transformer models, which has limited direct ethical concerns. The methods presented here aim to enhance mathematical problem-solving, primarily for academic and educational purposes. Since the model is designed for symbolic computation, the potential for misuse is minimal, and the outcomes are easily verifiable. As with all AI systems, it is important to ensure that results are used appropriately in domains requiring high mathematical rigor, but we see no significant ethical risks associated with this research.

## 11 REPRODUCIBILITY STATEMENT

To ensure the reproducibility of our results, we provide a detailed explanation of the methods and algorithms used, including the design of the symbolic engine, data generation pipeline, and model architecture. We also share all of the source code used to obtain the results. The codebase is well-structured, allowing researchers to replicate our experiments and build upon our work.

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

## A   SYMBOLIC ENGINE

### A.1   LIST OF ACTIONS IN THE SYMBOLIC ENGINE

Below, we present a complete list of actions available in the symbolic engine.

Table 4: List of Actions in the Symbolic Engine

| Action | Description |
| --- | --- |
| ConstantMethod | Apply when integrand is a constant. |
| PowerMethod | Apply power rule integration when integrand is $x^n$. |
| ExpMethod | Apply when integrand is of the form $a^x$. |
| ConstantTimesMethod | Factor out constants from integrand. |
| ReciprocalMethod | Apply when integrand is $\frac{1}{x}$. |
| NestedPowMethod | Handle integrals with nested powers. |
| ArcsinMethod | Apply when integrand is $\frac{1}{\sqrt{1-x^2}}$. |
| ArcsinhMethod | Apply when integrand is $\frac{1}{\sqrt{x^2+1}}$. |
| SinMethod | Apply when integrand is $\sin(x)$. |
| CosMethod | Apply when integrand is $\cos(x)$. |
| SecTanMethod | Apply when integrand is $\sec(x)\tan(x)$. |
| CscCotMethod | Apply when integrand is $\csc(x)\cot(x)$. |
| Sec2Method | Apply when integrand is $\sec^2(x)$. |
| Csc2Method | Apply when integrand is $\csc^2(x)$. |
| SinhMethod | Apply when integrand is $\sinh(x)$. |
| CoshMethod | Apply when integrand is $\cosh(x)$. |
| ArctanMethod | Apply when integrand is of the form $\frac{1}{ax^2+b}$. |
| ReciprocalSqrtQuadraticMethod | Apply when integrand is $\frac{1}{\sqrt{ax^2+bx+c}}$. |
| CiMethod | Apply when integrand is $\frac{\cos(ax+b)}{x}$. |
| EiMethod | Apply when integrand is $\frac{\exp(ax+b)}{x}$. |
| UpperGammaMethod | Apply when integrand is $x^n\exp(ax)$. |
| AddMethod | Rewrite integral of sum as sum of integrals. |
| UMethod | Substitute $u = f(x)$ and transform the integral. |
| PartsMethod | Apply integration by parts with parameters $u$ and $dv$. |
| PartialFractionsMethod | Decompose rational integrands into partial fractions. |
| CancelMethod | Simplify integrand by canceling terms. |
| ExpandMethod | Expand integrand algebraically. |
| Tan1Method | Rewrite $\tan(x)$ to $\frac{\sin(x)}{\cos(x)}$. |
| Cot1Method | Rewrite $\cot(x)$ to $\frac{\cos(x)}{\sin(x)}$. |
| Cos1Method | Rewrite $\frac{1}{\cos(x)}$ to $\sec(x)$. |
| Sec1Method | Rewrite $\sec(x)$ integrals using $\sec(x)^2$ and $\sec(x)\tan(x)$. |
| Csc1Method | Rewrite $\csc(x)$ integrals using $\csc(x)^2$ and $\csc(x)\cot(x)$. |

| Tanh1Method | Rewrite $\tanh(x)$ to $\frac{\sinh(x)}{\cosh(x)}$. |
|---|---|
| Coth1Method | Rewrite $\coth(x)$ to $\frac{\cosh(x)}{\sinh(x)}$. |
| Sech1Method | Rewrite $\mathrm{sech}(x)$ integrals using hyperbolic identities. |
| Csch1Method | Rewrite $\mathrm{csch}(x)$ integrals using hyperbolic identities. |
| TrigExpandMethod | Expand trigonometric functions in the integrand. |
| SinCosEvenMethod | Rewrites $\sin^m(x)\cos^n(x)$ as $(((1 - \cos(2ax))/2)^{m/2})((1 + \cos(2bx))/2)^{n/2}$ when $n$ and $m$ are even and nonnegative. |
| SinOddCosMethod | Rewrites $\sin^m(x)\cos^n(x)$ as $(1 - \cos^2(ax))^{(m-1)/2}\sin(ax)\cos^n(bx)$ when $m$ is odd and $m \geq 3$. |
| CosOddSinMethod | Rewrites $\sin^m(x)\cos^n(x)$ as $(1 - \sin^2(bx))^{(n-1)/2}\cos(bx)\sin^m(ax)$ when $n$ is odd and $n \geq 3$. |
| SecEvenTanMethod | Rewrites $\sec^n(x)\tan^m(x)$ as $(1 + \tan^2(bx))^{(n/2-1)}\sec^2(bx)\tan^m(ax)$ when $n \geq 4$ and $n$ is even. |
| TanOddSecMethod | Rewrites $\sec^n(x)\tan^m(x)$ as $(\sec^2(ax) - 1)^{(m-1)/2}\tan(ax)\sec^n(bx)$ when $m$ is odd. |
| Tan2Method | Rewrites $\tan^2(ax)$ as $\sec^2(ax) - 1$. |
| CotCscEvenMethod | Rewrites $\cot^m(x)\csc^n(x)$ as $(1 + \cot^2(bx))^{(n/2-1)}\csc^2(bx)\cot^m(ax)$ when $n \geq 4$ and $n$ is even. |
| CotOddCscMethod | Rewrites $\cot^m(x)\csc^n(x)$ as $(\csc^2(ax) - 1)^{(m-1)/2}\cot(ax)\csc^n(bx)$ when $m$ is odd. |

## B  REPRESENTATION OF MATHEMATICAL EXPRESSIONS

### B.1  ALGORITHM FOR SEQUENCE TO TREE

---
**Algorithm 1:** Function *seqtotree* to map from sequences to expression trees

**Data:** A sequence $seq$
**Result:** A tree $t$, and remaining part of the sequence $remaining$
*assert* $\mathrm{len}(seq) > 0$;
$t \leftarrow Node(seq[0])$;
**if** $t \in \{SYMBOL, CONSTANT, NUMBER\}$ **then**
   | **return** $parse(seq), remaining$ ;             /* parse greedily */
**else**
   | $a \leftarrow$ arity of the operand/function $t$;
   | $i \leftarrow 1$;
   | $remaining \leftarrow seq[1:]$;
   | **while** $i \leq a$ **do**
      | $child_i, remaining \leftarrow$ *seqtotree*$(remaining)$;
      | add $child_i$ to Node $t$;
   | **end**
   | **return** $t, remaining$
**end**

---

## C DATASET GENERATION

### C.1 GENERATING RANDOM MATHEMATICAL EXPRESSIONS

The algorithm for generating random mathematical expressions consist of three steps:

1. Sample random unary-binary trees with a uniformly distributed number of nodes between 3 and $N$, where $N = 50$.
2. Fill the leaves of the tree with a symbol $x$ with probability $p = \frac{3}{4}$ and one of the constants $\pi$, $e$ or number $\{0, \ldots, 10\}$ uniformly at random with probability $1 - p = \frac{1}{4}$. This is aimed at generating more difficult expressions effectively by incentivizing symbols as leaves.
3. Fill remaining internal nodes with random unary or binary operations. The binary operators are $+, -, \times, /$, and the unary operators include trigonometric, hyperbolic, their inverses, and the $exp$ and $log$ functions.

When selecting binary operations, addition and multiplication are twice as likely to be chosen over division and subtraction, as the latter can result in term cancellations and duplicate values. Unary operations are sampled uniformly at random. We do all trivial simplifications (e.g. evaluating $x + 2x$ to $3x$ or $x + 1 + 0 + 3$ to $x + 4$. All duplicates are removed during post-processing.

### C.2 DATASET STATISTICS

We report statistics for the final dataset in Table 5

Table 5: Dataset Statistics

| Expressions | Steps | Average Tokens | Average Steps | Min,Max Steps |
|---|---|---|---|---|
| 4.9M | 42.5M | 58.6 | 8.7 | 1,53 |

## D EXAMPLES OF SOLUTIONS GENERATED BY ALPHAINTEGRATOR

We show 2 more examples of solutions generated by AlphaIntegrator to demonstrate the tactics used by the model. First example requires a change of variables combined with understanding that the resulting integral is the non-elementary exponential integral, which AlphaIntegrator finds as first candidate in its search. Second example demonstrates that the model can handle long chains of computation where it has creatively find correct parameters for substitution and integration by parts.

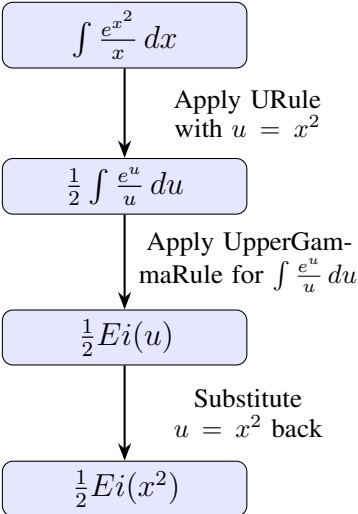

Figure 6: Step-by-step solution for $\int \frac{e^{x^2}}{x} \, dx$.

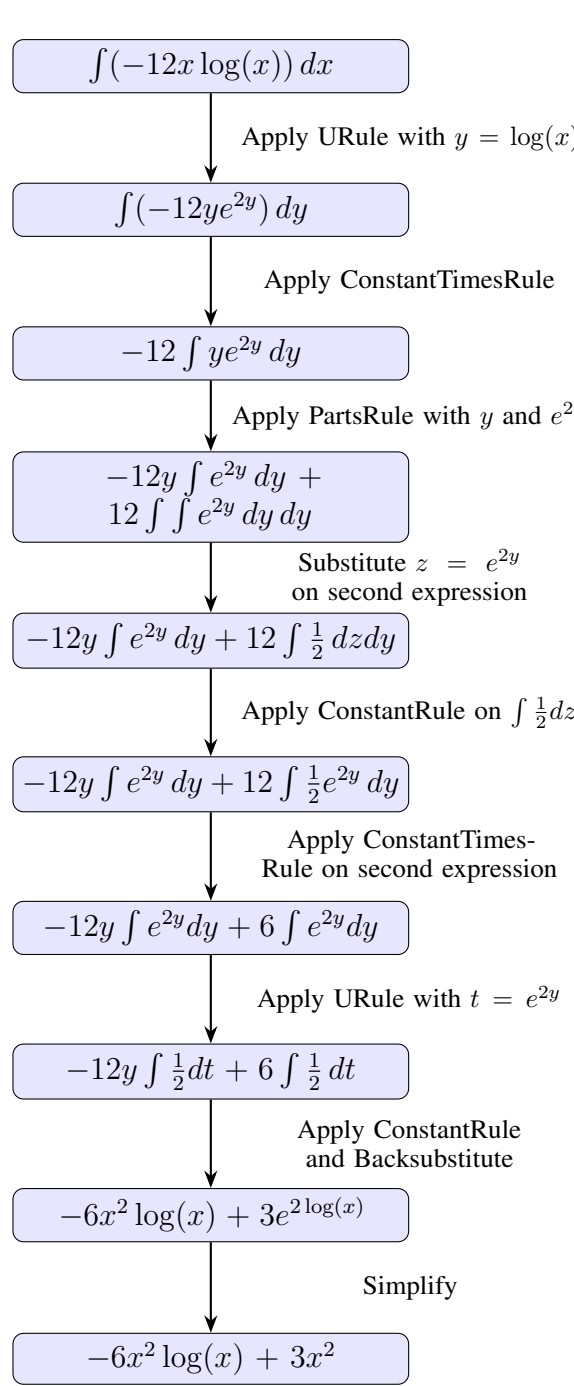

Figure 7: Step-by-step solution for $\int (-12x \log(x)) \, dx$.

