# OpenReview forum: "AlphaIntegrator: Transformer Action Search for Symbolic Integration Proofs"
_ICLR.cc/2025/Conference — Submitted to ICLR 2025_

### Official Review · Reviewer_QbVw · 2024-10-30

**Soundness:** 2
**Presentation:** 2
**Contribution:** 3
**Rating:** 6
**Confidence:** 3

**Summary:**

This paper presents a novel mathematical integration system that fuses neural models with symbolic solvers. In this approach, a GPT transformer model guides the selection of the appropriate integration rule, while a symbolic solver executes the precise calculations. To fine-tune the language models, the authors propose augmenting seed data to create a large-scale dataset of step-by-step integration. The experimental results demonstrate the proposed method's efficiency and comprehensiveness.

**Strengths:**

- Although the high-level insight is inspired by AlphaGeometry and AlphaProof, the application task and designed framework are very interesting and innovative.
- The authors provide the implementation of the proposed methods, offering valuable resources to the research community.
- The experimental results are promising, demonstrating that the proposed method significantly outperforms the baseline approaches.
- The differentiation analysis between the symbolic solvers and the proposed system could be useful for identifying bugs or limitations in existing heuristics.

**Weaknesses:**

- I am struggling to understand the practical motivation behind the research problem. First, the authors use SymPy as an example, which is notably inefficient compared with other symbolic tools. Second, I cannot see the necessity of employing an LLM is unclear, especially considering the model size is very small and the possible action space is well-defined and finite.
- The paper lacks comparisons with additional baselines. Although the proposed method is open-sourced, it would be helpful to include comparisons with proprietary systems, such as Wolfram's integral calculator, to serve as a point of reference.

**Questions:**

- The authors mention that SymPy's *manualintegrate* is slow and prone to failure. Could this be more of an implementation issue rather than an algorithmic one? For example, the SymPy is implemented fully on Python, causing low calculation efficiency.
- I am also curious about the time consumption. What is the average runtime for SymPy and the fine-tuned model on the test set? Additionally, is the model inference performed on CPUs?
- The data contamination problem should be further discussed. With the generation of approximately 45 million data points, there is a potential risk of overlap between training and testing datasets. Could the generated data lead to contamination issues, and how can these risks be mitigated during data generation?

---

> ### Author Response · Authors · 2024-11-20
>
> Thank you for your review.
>
> W1: The practical motivation of this paper is to improve existing open-source baselines on step by step theorem proving, and SymPy is the best existing alternative out there. The reason to use LLMs is that, on the contrary to your comment, action space is not finite. The model needs to pick exactly what subexpression to apply the action. In addition, the model needs to pick action parameters whenever it wants to apply the substitution rule or integration by parts. As the space of mathematical expressions is infinite, our action space is infinite.
>
> W2: Since Mathematica is closed-sourced, we found it difficult to make experiments at scale (with thousands of expressions) which is what we need to obtain an accurate number.
>
> Q1: As explained, the SymPy's algorithm works in a brute force manner, as you can see from the source code here: https://github.com/sympy/sympy/blob/master/sympy/integrals/manualintegrate.py
>
> Q2: We took data contamination seriously and made sure that integration steps that appear in the training set do not appear in the test set. However, integration is a problem that you can inherently just overfit as it is an algorithmic task (i.e. if the model discovered an almighty algorithm to integrate, it would be overfitting to the entire set of expressions in a way).
>
> We would be happy to answer if you have further questions.

---

> > ### Comment · Reviewer_QbVw · 2024-11-21
> >
> > Thank you for your response. I now have a better understanding of the motivation behind AlphaIntegrator, and as a result, I have increased my score.
> >
> > > Since Mathematica is closed-sourced, we found it difficult to make experiments at scale (with thousands of expressions) which is what we need to obtain an accurate number.
> >
> > Do you mean that Mathematica imposes limitations on the number of cores or processes that can be utilized? However, I may consider that an experiment on smaller test set or even on some typical cases could be helpful in demonstrating the strengths of AlphaIntegrator.

---

> > > ### Author Response · Authors · 2024-11-21
> > >
> > > Thank you very much for taking the time for this discussion.
> > >
> > > When we tried to make API calls we had some rate limit issues (it wouldn't allow us to make more API calls, we believe it's because they don't give away too much of their propietary algorithm). Furthermore, the system does not always give back a step-by-step solution which rendered the comparisons meaningless (for example, whenever it is exponential integral Ei, it just outputs the answer). This makes it very difficult to compare for the step-by-step task.

---

### Official Review · Reviewer_zJWe · 2024-11-02

**Soundness:** 2
**Presentation:** 3
**Contribution:** 1
**Rating:** 3
**Confidence:** 3

**Summary:**

This paper introduced AlphaIntegerator which trains a decoder-only transformer model to perform step-by-step integration. The expression to be integrated is represented by a mathematical expression tree and converted to a sequence in a depth-first manner. The model is then trained to predict single-step rules to be applied to the expression. Trained and evaluated on a synthetic dataset, the paper finds that AlphaIntegerator outperforms sympy and GPT-4o-mini.

**Strengths:**

The paper is clearly written and easy to follow.

The paper implements a learning approach to perform step-by-step integration.

**Weaknesses:**

The model’s ability for generalization could be limited, since the operators, variables, and actions are all represented as special tokens, and it doesnt seem like new operators, variables, or actions can be easily introduced without re-training the model. The model also currently only include 7 symbols to represent variables, which suggests that it cannot generalize to harder problems from the same distribution with more variables.

The experiments conducted are not extensive. Only the synthetic dataset is used and only 2 other methods are compared. Furthermore, the proposed method’s improvement from existing methods is not substantial (83.3% -> 87.3% accuracy).

Part of the synthetic data generation process is augmenting the dataset based on integration by parts, which later is shown to be a failure mode for the sympy solver. The comparison on the synthetic data may therefore favor the proposed model over sympy.

The motivation for this method is unclear to me. The paper states that it aims to “combine LLMs with symbolic reasoning engine”, but the model implemented is a 6-layer transformer trained from scratch.

**Questions:**

What were the design decisions behind including 7 symbols for variables? How easy is it for the model to handle more complex integration problems?

The model is required to generate subexpressions for integration, is the subexpression generated always valid? Is it being enforced?

Did you consider other methods for representing the expression other than using an expression tree + DFS? The transformer takes a sequence as input, can’t it handle the mathematical expression directly? (for example, INT+ 3 + x instead of + INT+ 3 x)

How does your work relate to existing work such as [1] combining symbolic solvers and LLMs?

Minor Comment:
The example in section 3.2 is a bit confusing to me, what does * INT+ 2 refer to in the expression?

[1] Pan, Liangming, et al. "Logic-lm: Empowering large language models with symbolic solvers for faithful logical reasoning.”

---

> ### Author Response · Authors · 2024-11-20
>
> Dear Reviewer,
>
> Thank you for your comments.
>
> 1. We believe you misunderstand what we mean by allowing 7 variables: these variables are used for applying the substitution rule such as u = sqrt(x^2 +1). We just had to set an upper limit as a token needs to be initialized for each such variable - this doesn't have anything to do with the capabilities of the model.
>
> 2. The generated subexpressions were almost always valid, but it is not necessary that this will be the case. We discard the proposed action whenever the subexpression is invalid.
>
> 3. Note that the proposed Polish notation is just another way of writing mathematical expression: representing expressions this way removes the redundancy due to parentheses. For example, the infix expression (3 + 4) * (5 - 2) can be written in Polish notation as * + 3 4 - 5 2, eliminating the need for parentheses and reducing token usage.
>
> 4. Regarding the example in Section 3.2, * INT+ 2 alone doesn't mean anything - you should rather look at it as * (param 1) (param 2) which represents multiplication of param1 and param2. In this case, we have * INT+ 2 POW cosh x INT+ 2 x which represents 2 * cosh^2(x).
>
> We don't really understand how training a 6-layer transformer from scratch is not compatible with the goal of "combining LLMs with symbolic reasoning engine." Note that the model we trained is a large language model - it doesn't have to be the case that it is pretrained or outputs natural language.
>
> We also see that you gave score of 1 for Soundness and 2 for Presentation but didn't have any comments on what are the problems with soundness and how the presentation could be improved. We would be happy to see your feedback so that we can improve our paper.

---

> > ### Author Response · Authors · 2024-11-21
> >
> > Dear Reviewer,
> >
> > We wanted to check in again to see if the reviewer has any questions about the paper. If not, we respectfully encourage the reviewer to revisit the scores in light of the clarifications. If any concerns remain, we would greatly appreciate additional feedback or suggestions for improvement.

---

> > ### Comment · Reviewer_zJWe · 2024-11-23
> >
> > Thank you for the response. Some further comments:
> >
> > >  7 variables are used for applying the substitution rule. Doesn't have anything to do with the capabilities of the model.
> >
> > Does this mean a single solution to a problem can only apply the substitution rule 7 times? If so, this limits the capabilities of the model in the sense that it can not generalize to harder problems requiring more substitution steps. If not, can you explain the design choice of using 7?
> >
> > > We discard the proposed action whenever the subexpression is invalid.
> >
> > How does the model check whether a subexpression is invalid?
> >
> > > representing expressions this way removes the redundancy due to parentheses.
> >
> > I see, that makes sense.
> >
> > > The model we trained is a large language model, it doesn't have to be the case that it is pretrained or outputs natural language.
> >
> > While the transformer architecture is the backbone of LLMs, the term LLM specifically refers to large-sized pre-trained language models (billions of parameters) [1].
> >
> > [1] Zhao, Wayne Xin, et al. "A survey of large language models." arXiv preprint arXiv:2303.18223 (2023).
> >
> > > \* INT+ 2 POW cosh x INT+ 2 x represents 2 * cosh^2(x).
> >
> > Thanks for the clarification, my confusion stemmed from the expression in the paper showing cosh^2(x) instead of 2 * cosh^2(x) which I believe is a typo.
> >
> > > Soundness and Presentation
> >
> > I have adjusted these scores accordingly. However, I am still recommending reject based on the points I discussed in the weaknesses section.

---

> > > ### Author Response · Authors · 2024-11-23
> > >
> > > Dear reviewer,
> > >
> > > 1. Yes, that is correct indeed. We had to set an upper bound to this - and it was already the case that 7 variables were more than enough. Note that some variables free up once they are backsubstituted so the symbolic engine can do more than 7 substitutions without needing 7 different variables.
> > >
> > > 2. The transformer model does not check that the subexpression is valid: this is done through the symbolic engine. We simply try to parse a tree, for which we know the formal grammar, so we can check whether it is valid or not. Note that the model almost always (>%99) generates valid mathematical expressions.
> > >
> > > 3. We disagree with your criticism that %4 improvement is not substantial - we would like to remind that it is typically much more difficult to improve the last few percentage points than the first. We furthermore demonstrate that our model explores x2 less nodes than SymPy which is a very strong improvement to the existing solver.
> > >
> > > `Part of the synthetic data generation process is augmenting the dataset based on integration by parts, which later is shown to be a failure mode for the sympy solver. The comparison on the synthetic data may therefore favor the proposed model over sympy.`
> > >
> > > Note that the models were compared on the test dataset generated from random expressions, not on the augmented dataset. Hence we disagree with this criticism.
> > >
> > > To clarify, the motivation behind this research is to develop a neurosymbolic system that can 'provably' solve mathematical integration, rather than simply guessing the answer. We show that this method outperforms existing methods which gives a strong signal that the same approach can be applied to many other tasks, which we are looking forward to do in our further research.
> > >
> > > We hope this clarifies the points you raised in the weakness section and we're looking forward to your further feedback.

---

### Official Review · Reviewer_jvis · 2024-11-04

**Soundness:** 2
**Presentation:** 2
**Contribution:** 2
**Rating:** 5
**Confidence:** 3

**Summary:**

This paper introduces a correct-by-construction learning-based system for step-by-step mathematical integration. Concretely. LLMs interact with a symbolic engine with a set of handcraft rules. This paper also contributes a dataset for rigorous step-by-step derivation proofs for indefinite integration. The proposed model is trained and tested on the synthesized dataset and compared with two baselines, SymPy and GPT-4o-mini.

**Strengths:**

* The paper introduces a new approach to integrating LLMs with symbolic reasoning engines for mathematical integration.
* The paper is clearly written and easy to follow.

**Weaknesses:**

Please see the questions below.

**Questions:**

* The representation of mathematical expressions and theorems are trees consisting of operands/functions with fixed arities. For example, addition and multiplication only have two children. Therefore, the proposed method seems challenging to extend to general situations.
* The data synthesis results in 4.9 expressions. What is the diversity of the synthetic expressions? Are there specific types of expressions that are underrepresented in the training data, and how might this affect the model's performance?
* While the paper compares AlphaIntegrator with SymPy and GPT-4o-mini, it would be informative to see a comparison with other state-of-the-art symbolic solvers or theorem provers.
* Are there specific types of integrals or mathematical expressions where AlphaIntegrator excels or falls short compared to these tools?
* How does AlphaIntegrator perform on open-source math datasets?
* The listed references are not formal. For example, 'Deep Learning For Symbolic Mathematics' has already been published in ICLR, but the citation is still CoRR.

---

> ### Author Response · Authors · 2024-11-25
>
> Dear Reviewer,
>
> We hope our general rebuttal answered most of your questions. We wanted to clarify a few more points specific to your questions as well:
>
> 1. Note that the only requirement is that an operation has "fixed" arity, but the same model is generalizable to expressions of any arity.
>
> 2. Existing theorem provers are usually unable to carry out this kind of computations to solve integrals, as they're more focused on mathematical theorem proving in fields such as number theory and algebra. Likewise, symbolic solvers do not necessarily output steps as we discussed in our general rebuttal.
>
> 3. In our evaluation, we find that AlphaIntegrator performs strictly better than the existing step by step solvers. However, solving integration step by step is not always possible, in which cases the model is unable to get to such solution.
>
> 4. Note that the integrals generated by are a lot more difficult than the standard textbook problems - our model is able to successfully solve most problems (>%95) in standard calculus textbooks. We promise to include more detailed analysis in the final version of the paper.
>
> 5. Thanks for this feedback - we will fix the citation in the final version.
>
> We hope this clarifies your concerns and we kindly invite you to update your scores!

---

### Official Review · Reviewer_1cA8 · 2024-11-04

**Soundness:** 3
**Presentation:** 4
**Contribution:** 2
**Rating:** 5
**Confidence:** 4

**Summary:**

The paper presents AlphaIntegrator -- a learning-based system for computing antiderivatives of mathematical expressions in a step-by-step manner. The system is a hybrid of a symbolic engine checking for correctness of each step, predefined symbolic "actions", and a transformer-based policy pretrained on a sizable set of synthetic examples.

The authors evaluate their system on a synthetic dataset and show its superiority compared to the symbolic approach implemented in SymPy, as well as GPT-4o-mini model.

**Strengths:**

1. The paper tackles an important and interesting problem of synergizing learning-based and formal/symbolic methods.
2. AlphaIntegrator is shown to perform better that GPT-4o-mini and SymPy. Additionally, it is demonstrated that AlphaIntegrator is less "symbolically brittle" than the approach by Lample and Charton (2019) computing antiderivatives directly.
3. As a byproduct of the project, the authors contributed a few fixes to SymPy's integration algorithm.
4. The paper is very clearly written, providing enough background and helpful examples.
5. The authors provide the code for reproducing the experiments (the code looks runnable, although I didn't test it.).

**Weaknesses:**

1. There is no comparison of AlphaIntegrator with the approach by Lample and Charton (2019). I find it a serious omission.
2. There is no comparison of AlphaIntegrator with the integration algorithm in Mathematica (which, I think, is stronger than this implemented in SymPy).
3. The achieved improvement is rather small compared to SymPy's performance.
4. The proposed approach is not super novel: there is already quite a large body of research dealing with ML-based formal theorem proving, where the setting is similar as in AlphaIntegrator: an ML model suggests actions (proof steps, aka "tactics") constrained by the formal/symbolic environment.
5. The target problem -- integration -- seems somewhat not very well suited for that kind of approach. I suppose that the purely symbolic approaches deal with most of the integration problems efficiently. Moreover, it is not crucial to have a step-by-step derivation of the antiderivative: once we have a antiderivative, verifying its correctness is easy, and step-by-step solution is irrelevant.
6. The authors claim that their approach can be adapted to other kinds of math problems, but I find this claim unjustified.
7. The authors did not release the test dataset, which makes it more difficult to compare with AlphaIntegrator in the future. (Or did you release? I'm not sure here.)
8. The authors instantiate the learning component of AlphaIntegrator with a transformer model, and do not test any other ML approaches.

**Questions:**

1. Am I right that you input the expressions to the transformer using prefix Polish notation? Why did you prefer this notation over the standard one? Did you run an experiment justifying this choice?
2. Did you generate the synthetic dataset yourself or reused the one provided by Lample and Charton (2019)? You mention that you generate it in line 193, but in the provided code I see you download the data of Lample and Charton.
3. Where do the predefined actions come from? Did you experiment with different sets of actions?
4. Did you experiment with other types of tree search? For instance, breadth-first search instead of best-first search induced by the log-probs?
5. To what extent the set of solutions by AlphaIntegrator overlaps with the solutions by SymPy?
6. Do you think TreeNN architectures would be better suitable as the learning component of AlphaIntegrator? Or maybe even simpler ML approaches or heuristics would be applicable given the predefined list of actions? (Like k-NN?)
7. You write "We run SymPy and our own model with timeouts of 120 and 10 seconds, respectively." Why different timeouts? Is that correct that SymPy got more time than AlphaIntegrator?
8. How and to what other math problems can AlphaIntegrator be adapted?
8. How did you split the data into training and testing parts? Could you release both datasets for inspection?

---

> ### Author Response · Authors · 2024-11-21
>
> Thank you for the detailed review.
>
> Regarding the weaknesses pointed out by the reviewer:
>
> 1. Comparison to Lample and Charton (2019). This paper solves a different problem (we generate correct steps that may or may not lead to a full solution, they generate a possibly incorrect guess for the antiderivative). However, as we used the same datasets of integration problems, our numbers are directly comparable to the extent that that is meaningful.
>
> 2. Comparison to Mathematica: It is tricky to compare at scale to a baseline that is not open source. It seems more fruitful for Wolfram to eventually adopt an approach like ours to improve Mathematica. We would of course have done this instead if it were open, but it seems unfair to expect us to compete with a for-profit organization that is not bound by basic best practices for computer science research. We are however also curious and will see at what level such a comparison is possible (e.g. by subsampling the test set).
>
> 3. Only small improvement over SymPy: We disagree with this point. Given that all the training data stems from SymPy, which the model is instructed to imitate, we think that beating it by 4% while exploring only half of the search space on average, with fewer outliers, is an impressive result. We did not expect the model to outperform the training data.
>
> 4. Novelty: The main novel contribution is the specific action space, action encoding, implementation, and the evaluation. What we propose is certainly not a paradigm shift in symbolic reasoning, but we think it would be unreasonable to expect every publication that generates correct-by-construction results to deviate from the obvious "actions in symbolic environment" high-level setting in a non-trivial manner.
>
> 5. Integration steps not crucial to obtain final result, as it is easy to verify: Verifying the final result is an algorithmically undecidable problem, equivalent to the zero problem. We however agree that it is often easy in practice. The step-by-step approach is interesting when reasoning steps are actually important (it is not a coincidence that wolframalpha.org puts reasoning steps behind a paywall), and it enables more interesting applications down the line, such as non-trivial combination of multiple action spaces that operate on the same underlying representation. Also, AlphaIntegrator beats the open source symbolic step-by-step baseline by a respectable margin.
>
> 6. Adaptibility to other domains not justified. We make two references to adaptability. One is that transformer models are more versatile than TreeNN, which we think does not require additional justification. The other one is in future work, and therefore indeed speculative.
>
> 7. Test dataset not released: We released all code that is required to run our evaluation. However, that indeed does not include the specific test dataset as there is some randomness in how it is derived from Lample and Charton, due to shuffling. We did not run with a fixed seed. We will release the full training and test sets that we used in our evaluation, though we think the standard for reproducibility should be that new runs from scratch still produce results compatible with the mean and error margin reported in our paper.

---

> ### Author Response · Authors · 2024-11-21
>
> We are happy to address the questions:
>
> 1. Polish notation: Our first experiments indicated that the Polish notation indeed makes training easier for our tasks which is why we decided to stick with it.
>
> 2. Synthetic dataset generation: Thank you for pointing out this discrepancy! We indeed used the data of Lample and Charton (2019) directly and we will make this clear in the next version of the paper.
>
> 3. Predefined actions: There is a lot of freedom around how one can choose actions for such a system: we used two critera to design the action space. First, it had to be comprehensive enough to cover a wide range of integrals. Second, it had to be suitable for converting the steps generated by SymPy. We did not experiment with another action space than the one that we present in the paper.
>
> 4. Comparison to other types of tree search: The SymPy baseline uses a different type of tree search without log-probabilities and expores more nodes while solving fewer integrals. One might indeed also consider what happens when one uses a non-guided search method with actions generated from the model, but it appears that would be strictly worse than the SymPy baseline, so we had not considered such an experiment. In general, experimenting with different types of tree search is an interesting direction for future work.
>
> 5. AlphaIntegrator vs SymPy overlap: For integration problems solved by both approaches, there is ~98% overlap in actions taken. We will add this statistic to the paper.
>
> 6. TreeNNs: Maybe. We have not tried this so far, but it would be an interesting comparison to make. However, our approach is more flexible and we demonstrated that it works well _even though_ we did not make any specific accommodations for the structure of the input data.
>    Simpler ML heuristics, like k-NN: That is an interesting suggestion for a baseline. However, it is not so clear to us how to make k-NN work in our setting. E.g., how to compute distances between integration states? It seems that ranking all the applicable actions that are present in the training dataset is itself a tricky problem, and even just testing all of those actions for applicability in each step does not seem feasible.
>
> 7. Higher SymPy timeouts: Yes, it is correct that the SymPy baseline timeout was significantly higher. This is to make up for the inefficient implementation of SymPy, as our goal was to show that SymPy cannot solve these integrals rather than to show that the Python interpreter is slow. We will increase the time limit also for our own approach and report a more quantitative analysis of the running times (e.g., a plot of integrals solved over time for the different approaches).
>
> 8. Adapting to other approaches: AlphaIntegrator demonstrates that our way of applying actions to subtrees is effective in this setting. The approach is in principle generalizable to any step-by-step reasoning task. However, significant effort is required to identify a suitable action space in those different contexts, particularly because we need to then be able to generate a data set that demonstrates how to effectively apply these actions. So while we think exploring other applications of the same kind of method is promising, we think such follow-up work would likely still require interesting research.
>
> 9. Training/test data: We used the original split from Lample and Charton (2019). In the interest of full transparency and reproducibility, we are happy to upload the final datasets derived from Lample and Charton that we used for training and testing.

---

> > ### Author Response · Authors · 2024-11-24
> >
> > Dear Reviewer,
> >
> > We hope that our rebuttal answered your questions and concerns. We kindly invite you to update your scores and we're happy to answer if you have any further questions.

---

> > > ### Comment · Reviewer_1cA8 · 2024-12-03
> > >
> > > Thank you for your rebuttal. However, I remain sceptical about the setting and research goal of the project, and this is why I keep my score below the acceptance threshold. I think that demonstrating additional results for some more challenging problem (like multivariable integrals, or solving differential equations?) would strengthen the paper.

---

### Author Response · Authors · 2024-11-20
**General Response**

Dear reviewers,

We greatly appreciate all of your feedback. As there are a lot of overlapping questions, we decided to reply to all in this comment.

1. We did not explicitly compare with Lample and Charton (2019) as it does not provide steps to solve the integral but only the antiderivative itself. The focus our work is *step-by-step* integration, as this generalizes well beyond finding the antiderivative. In fact, you can only do the antiderivative check for single variable integrals: multivariable integrals over complex domains require step by step solution.

2. Some reviewers claim that improvement is minimal, however we disagree that %4 improvement is small, given that the model was trained on synthetic data generated by SymPy.

3. Symbolic approaches indeed work quite well on the task of finding the antivderivative. - however they rarely give step by step solutions which is important in many cases. To give a concrete example, a similar methodology can be applied to solving multidimensional integrals where the domain is non-standard or rewriting inequalities in a desired way. This is a great example on how our model can be extended to different problems as well.

4. For the test dataset, we used the same one from Lample and Charton to ease up comparisions. For the dataset of random expressions, we used the dataset generated by Lample and Charton, however, we generated the data for the step by step solutions. This datasets can be found on the GitHub page of the Lample and Charton paper: https://github.com/facebookresearch/SymbolicMathematics

5. Our first experiments indicated that the Polish notation indeed makes training easier for our tasks which is why we decided to stick with it.

6. There is a lot of freedom around how one can choose actions for such a system: we used 2 criterion to design the action space. First, it had to be comprehensive enough to cover a wide range of integrals. Second, it had to be suitable for converting the steps generated by SymPy.

7. We did not have time to try different search methods, but we believe that our results could only further improve with better search methods - we will work on such improvements in the next steps of the project.

8. We ran SymPy with a longer timeout to make sure that we get its saturated performance that is not affected by the slow execution time of Python.

We hope this answers your questions and we're looking forward to your further feedback.

---

> ### Comment · Reviewer_1cA8 · 2024-11-21
>
> Thank you for the comment. I have some further questions:
>
> You write
>
> > We did not explicitly compare with Lample and Charton
>
> but also
>
> > For the test dataset, we used the same one from Lample and Charton to ease up comparisions.
>
> So why didn't you show and discuss the comparison with Lample and Charton in the paper when your setup was designed to facilitate such a comparison? I understand that your method has the advantage of providing the step-by-step solution whereas Lample and Charton's method does not, but it would still be informative to compare the numbers.
>
>
>
> > you can only do the antiderivative check for single variable integrals: multivariable integrals over complex domains require step by step solution.
>
> That seems to be a good point, but in that case you could make your paper more convincing by showing experiments with multivariable integrals. In its current state, the paper only studies expressions for which having the step-by-step solution is not crucially important.
>
> > Our first experiments indicated that the Polish notation indeed makes training easier for our tasks which is why we decided to stick with it.
>
> Do you have some numerical results supporting this claim? Such an insight is valuable in itself, but we would need some evidence supporting it.

---

> > ### Author Response · Authors · 2024-11-21
> >
> > Thank you for your comment.
> >
> > 1. We didn't explicitly draw comparisions as the step-by-step solution problem is very different in nature compared to predicting the anti-derivative. We wanted to use the same dataset that is commonly used in other papers (for example the Symbolic Brittleness paper https://arxiv.org/abs/2109.13986) so that other researchers can easily benchmark. For completeness, we will include the results from Lample and Charton in the final version. In addition, we would like to emphasize that the paper "Symbolic Brittleness in Sequence Models: on Systematic Generalization in Symbolic Mathematics" demonstrated that seq2seq models like the ones from Lample and Charton are not robust to small changes in input, whereas we demonstrate that our model generalizes well through the robustness tests from this paper in Section 7.3.
> >
> > 2. The reason we didn't run experiments with multidimensional integrals is that it is a lot more difficult to generate synthetic data for this task due to lack of existing solvers that output steps in a format that is easily parsable - we're currently in the process of generating data for this from exact probabilistic inference engines such as PSI (https://www.semanticscholar.org/paper/PSI%3A-Exact-Symbolic-Inference-for-Probabilistic-Gehr-Misailovic/9b09341d63284351e80c95d3a80c4c241a3d8ed5) however these languages are not designed to generate this kind of data and hence it is non-trivial to extract relevant data for this problem. We would like to emphasize that our model also demonstrates that one can find bugs in software using our method through generalization capabilities of transformers as demonstrated in Section 7.4.
> >
> > 3. Unfortunately, we didn't store the full results during our experiments but the performance gap for the same number of epochs was statistically significant to fix using the Polish notation. Note that use of this notation is already very common in the literature due to the fact that it allows one to represent the same mathematical expression with significantly less tokens. We promise to add full experiments in the final version to demonstrate this gap between different representations.
> >
> > We hope this answers your questions and we respectfully encourage the reviewer to revisit the scores in light of the clarifications. If any concerns remain, we would greatly appreciate additional feedback or suggestions for improvement.

---

> > > ### Comment · Reviewer_1cA8 · 2024-11-22
> > >
> > > Thank you for clarifications. You write
> > >
> > > >  this notation is already very common in the literature
> > >
> > > Could you provide some references?

---

> > > > ### Author Response · Authors · 2024-11-22
> > > > **Examples of Polish notation**
> > > >
> > > > For example, you can check out the following work:
> > > >
> > > > 1. The paper that we widely cited: Lample and Charton (2019) (https://arxiv.org/abs/1912.01412) uses Polish notation to represent expressions and ODEs.
> > > > 2. End-to-end Symbolic Regression with Transformers (https://proceedings.neurips.cc/paper_files/paper/2022/hash/42eb37cdbefd7abae0835f4b67548c39-Abstract-Conference.html) from NeurIPS 2022.
> > > > 3. Piotrowski, Brown, Urban, Kaliszyk: Can Neural Networks Learn Symbolic Rewriting?, AITP 2019, http://aitp-conference.org/2019/aitp19-proceedings.pdf
> > > >
> > > > Thanks!

---

> > > > > ### Comment · Reviewer_1cA8 · 2024-12-03
> > > > >
> > > > > Thank you for those references, however, I could not find in them a comparison of the notations in terms of the performance. I think that it would be nice to see an experiment confirming that the Polish notation performs better -- it is really not obvious for me if that's the case.

---

### Author Response · Authors · 2024-11-27

We thank the reviewers again for their initial feedback. As we have not yet received replies to our rebuttal and the discussion period is closing shortly, we would like to kindly ask the reviewers to let us know if our responses addressed their concerns, and raise any follow-up questions.

---

### Meta-Review · Area_Chair_G7Re · 2024-12-23

**Metareview:**

This paper presents, AlphaIntegrator, which is a decoder-only transformer model trained to predict stepwise integration rules for computing antiderivatives of mathematical expressions. Groundtruth integration rules are collected by running SymPy over a synthetic dataset collected by the previous work Lample and Charton (2019). The experimental evaluation shows that AlphaIntegrator (combined with a symbolic engine) outperforms the baseline SymPy and GPT-4o-mini.

The general idea of using a language model to guide the search of an underlying symbolic engine is not new, but the specific application to math expression integration is new. The main concerns shared by reviewers and the AC are twofolds: a limited experimental evaluation setup and several unjustified claims. The authors emphasize that the new approach gives step-by-step integration rules which are further validated by the underlying symbolic engine, while many existing approach does not, which is indeed a nice property (or byproduct) of the proposed approach (or more broadly any ML-guided theorem proving approaches). However, simply excluding important baselines like the previous work Lample and Charton (2019) is problematic. Applying the same criteria, GPT-4o-mini should be excluded as well. Secondly, whether Polish notation is superior to other representations is vaguely hinted but not backed up with quantitative results. Other claims like dataset generation (line 193) turn out to be factually false.

**Additional Comments On Reviewer Discussion:**

The authors and reviewers had active discussions centering around Polish notation, synthetic datasets, important related works and baselines, action rules, and generalization capabilities. Many confusions regarding the notation, datasets, and action rules have been carefully clarified. There are disagreement about whether 4% improvement over the baseline SymPy is significant. The limited evaluation setup and lack of support from reviewers suggests the current work is not yet ready for publication.

---

### Decision · Program_Chairs · 2025-01-22

Reject